# Negative Affectivity Is Associated with a Higher Systolic and Diastolic Blood Pressure in Normotensive Middle-Aged and Older Adults: A Cross-Sectional Study

**DOI:** 10.3390/medicina56040160

**Published:** 2020-04-02

**Authors:** Tin-Kwang Lin, Chin-Lon Lin, Yi-Da Li, Ying Huang, Malcolm Koo, Chia-Ying Weng

**Affiliations:** 1Division of Cardiology, Department of Internal Medicine, Dalin Tzu Chi Hospital, Buddhist Tzu Chi Medical Foundation, Dalin, Chiayi 62247, Taiwan; 2School of Medicine, Tzu Chi University, Hualien City, Hualien 97004, Taiwan; 3Department of Psychology, National Cheng Chung University, Minxiong, Chiayi 62102, Taiwan; 4Graduate Institute of Long-term Care, Tzu Chi University of Science and Technology, Hualien City, Hualien 97005, Taiwan; 5Dalla Lana School of Public Health, University of Toronto, Toronto, M5T 3M7 ON, Canada

**Keywords:** blood pressure, hypertension, Type D personality, negative affectivity, social inhibition

## Abstract

*Background and objectives*: Type D personality, characterized by two stable traits (social inhibition and negative affectivity), is associated with adverse cardiovascular events. A possible mediating factor for this association could be hypertension. Previous research has shown that individuals with Type D personality were associated with an increased risk of hypertension. However, the association of negative affectivity and social inhibition on blood pressure in normotensive individuals has not yet been reported. Therefore, the aim of this study was to investigate whether negative affectivity and social inhibition were associated with systolic and diastolic blood pressure in normotensive middle-aged and older Taiwanese adults. *Materials and Methods*: A cross-sectional study design was used. Individuals attending general health examination at a regional hospital in southern Taiwan who were 40 to 75 years old were recruited. Patients with self-reported hypertension or currently receiving antihypertensive medication were excluded. Negative affectivity and social inhibition were assessed with the 14-item Type D Scale-Taiwanese version. Multiple linear regression analyses were conducted to determine the association of Z-score transformed negative affectivity and social inhibition on blood pressure. *Results*: A total of 92 patients with a mean age of 51.5 years were included in the study, and 15 (16.3%) were defined as having a Type D personality. The Z-score transformed negative affectivity score (*p* = 0.035, effect size = 0.18) and Z-score transformed social inhibition score (*p* = 0.054, effect size = 0.17) were significantly associated with a higher systolic blood pressure. In addition, the Z-score transformed negative affectivity score (*p* = 0.036, effect size = 0.28) and Z-score transformed social inhibition score (*p* = 0.154, effect size = 0.24) were significantly associated with a higher diastolic blood pressure. *Conclusions*: Negative affectivity of the Type D personality was significantly associated with higher systolic and diastolic blood pressure, with a medium effect size, in apparently healthy middle-aged and older adults. Assessment of negative affectivity may be clinically useful in identifying individuals at risk of hypertension.

## 1. Introduction

A distressed personality type or Type D personality is characterized by the joint tendency to experience negative emotions towards themselves and others, and to inhibit these emotions while avoiding social interaction. These dimensions are referred to as negative affectivity (NA) and social inhibition (SI), respectively [1]. The prevalence of Type D personality is estimated to range from 13% to 31% in the general population [2,3]. Type D personality has been shown to be associated with adverse outcomes in patients with coronary heart disease [4], cardiovascular disease [5], cognitive dysfunction [6], diabetes [7], metabolic syndrome [8], and depression [9]. Our previous study showed that both NA and SI were significantly associated with a higher risk of cardiac readmission in both 6 months and 18 months after the initial hospitalization in patients with heart failure [10]. A cross-sectional study on patients with coronary artery disease found that SI and NA were associated with the severity of the disease, which could be explained by the association between SI and poor health behavior, as well as a greater stress hormone reactivity associated with NA [11].

Most previous research on Type D personality has focused on patients with various diseases rather than on healthy individuals. For example, Type D personality was found to be significantly associated with hypertension, after adjusting for sex, age, body mass index, family history of hypertension, living condition, education, and employment [12]. However, the relationship between the components of Type D personality and blood pressure in normotensive adults has not yet been reported. Therefore, this study sought to investigate whether NA and SI were associated with systolic and diastolic blood pressure in normotensive middle-aged and older Taiwanese adults.

## 2. Materials and Methods

### 2.1. Study Design and Participants

The study protocol was approved by the Institutional Review Board of Dalin Tzu Chi Hospital, Buddhist Tzu Chi Medical Foundation (No. B09704030-1, approval date 27 May 2010). Informed consent was obtained from all participants in the study.

A cross-sectional study design was used to recruit individuals undergoing a general physical examination at a regional hospital in southern Taiwan from November 2009 to June 2010. Patients aged 40 to 75 years old were included in the study. Exclusion criteria included patients with self-reported hypertension or those who were currently receiving antihypertensive medication. In addition, patients with cardiovascular disease, stroke, and cancer were also excluded from the study.

### 2.2. Measurement of Type D Personality

The 14-item Type D Scale-Taiwanese version (DS14-T) [13] was used to assess the NA and SI of the participants. The original DS14 instrument was developed by Denollet and associates [2]. It consists of two 7-item subscales for measuring NA and SI, and the items were measured with a five-point Likert scale that ranges from 0 (false) to 4 (true). A standard cut-off score of ≥10 on both subscales is used to define Type D personality. The original scale showed good psychometric properties, including internal reliability with Cronbach alphas of 0.88 for NA and 0.86 for SI, and 3-month test–retest reliability with correlation coefficients of 0.72 for NA and 0.82 for SI [2]. The DS14-T version also demonstrated good internal consistency (Cronbach α = 0.86 and 0.79), with factor analyses that confirmed the two-factor model of the Type D construct [13].

### 2.3. Measurement of the Main Outcome and Other Covariates

Information of the patients, including age, sex, body weight, body height, educational level, marital status, family history of hypertension, exercise, smoking habit, and alcohol use were obtained from a questionnaire distributed at the start of the general health examination. Body mass index was calculated by dividing body weight in kilograms by body height in meters squared. Resting systolic and diastolic blood pressure were measured by trained staff using a vTrust 701DH digital handheld non-invasive blood pressure monitor (BioCare Corp., Taoyuan City, Taiwan), with patients in a seated position.

### 2.4. Statistical Analysis

Continuous variables were expressed as means with standard deviations, and categorical variables were expressed as numbers with percentages. Independent t-test and Chi-square test or Fisher’s exact test were used to compare patients with or without Type D personality. To aid the interpretation of the results for NA and SI, their values were transformed into Z-scores with a mean of zero and a standard deviation (SD) of 1. Simple and stepwise multiple linear regression analyses were conducted to determine the association between blood pressure and independent variables, including Z-score transformed NA and SI, age, sex, body mass index, educational level, marital status, family history of hypertension, regular exercise, current smoking, and current alcohol use. Effect size of regression (f^2^) was calculated by r^2^/(1 − r^2^). Values of f^2^ near 0.02, 0.15, and 0.35 could be defined as small, medium, and large, respectively [14]. All analyses were performed with IBM SPSS Statistics for Windows, Version 24.0 (IBM Corp., Armonk, NY, USA). A *p* value of <0.05 was considered statistically significant.

## 3. Results

A total of 171 patients with age ranging from 40 to 75 years were identified. Of those, 18 with cardiovascular disease, one with ischemic stroke, and seven with cancer were excluded. Of the remaining 149 patients, 92 (61.7%) reported that they did not have hypertension or were not receiving antihypertensive medications. These 92 patients were included in subsequent analyses.

The basic characteristics of the study participants, divided into Type D and non-Type D, are shown in Table 1. Among the 92 patients, 15 (16.3%) were defined as having a Type D personality, with a significantly higher proportion being female. In addition, the mean age and body mass index were lower in patients with Type D personality.

Table 2 shows the results of simple linear regression analyses of systolic and diastolic blood pressure. Only age was significantly associated with a higher systolic blood pressure (standardized [std.] β = 0.195, *p* = 0.025). For diastolic blood pressure, body mass index (std. β = 0.294, *p* = 0.004) and family history of hypertension (std. β = 0.215, *p* = 0.044) were significant factors. In addition, current smoking was marginally associated with a higher diastolic blood pressure. Both Z-score transformed NA and SI scores were not significantly associated with systolic or diastolic blood pressure. However, as shown in Table 3, the Z-score transformed NA score was significantly associated with systolic blood pressure (std. β = 0.220, *p* = 0.035) when adjusting for other significant covariates in the regression model (age and body mass index). It was also significantly associated with diastolic blood pressure (std. β = 0.218, *p* = 0.036) when adjusting for other significant covariates in the regression model (age, body mass index, and family history of hypertension). Conversely, the Z-score transformed SI score was associated with systolic blood pressure with only a marginal significance (std. β = 0.196, *p* = 0.054), and not significantly associated with diastolic blood pressure (std. β = 0.144, *p* = 0.154) in the regression models. The f^2^ for all four models were above 0.15, which could be interpreted as a medium effect size.

Age, sex, body mass index, educational level, marital status, family history of hypertension, regular exercise, current smoking, and current alcohol use were evaluated in the stepwise multiple linear regression model development. The Z-score transformed negative affectivity score and social inhibition score were forced to retain in their respective model.

Age and body mass index were other significant covariates in the systolic blood pressure model, whereas age, body mass index, and family history of hypertension were other significant covariates in the diastolic blood pressure model.

## 4. Discussion

This cross-sectional study showed that NA, a tendency to experience negative emotions, was significantly associated with systolic and diastolic blood pressure in normotensive middle-aged and older Taiwanese adults. To the best of our knowledge, this is the first study to report this association. Previous research generally focused on comparing Type D personality between hypertensive patients and healthy individuals. Denollet [2] reported that Type D personality was more prevalent in patients with hypertension (53%) as compared to healthy individuals (19%). A recent cross-sectional study of patients visiting general practitioners showed that Type D personality was associated with a 2.5-fold increase in the likelihood of hypertension [12]. Another cross-sectional study in the Icelandic population also reported that type D personality was associated with a significantly higher prevalence of hypertension [15]. On the other hand, the present study explored whether Type D personality was associated with blood pressure in individuals who were not diagnosed with hypertension. This is an important finding from a preventive point of view because educational and counselling interventions may be considered in normotensive individuals with Type D personality to reduce the risk of transition to hypertension.

Regarding the mechanism of how type D personality could affect blood pressure, Kupper et al. suggested that type D personality was associated with exaggerated α-adrenergic vasoconstriction to the cold pressor test, which might increase the risk of hypertension later in life [16]. In our previous study of patients with hypertension, those with a type D personality had significantly higher adjusted mean systolic and diastolic blood pressure at the recovery phase of anger recall, which suggested a prolonged blood pressure recovery in these individuals [17]. Another possible pathway through which Type D personality could exert an adverse effect on blood pressure was that these individuals might be less likely to engage in optimal health behavior. A British study of 564 adults reported that Type D personality was significantly associated with a sedentary lifestyle [18]. In addition, individuals with Type D personality were found to spend less time outdoors regularly, to eat less sensibly, and to receive fewer regular medical checkups compared with non-Type D individuals [19]. Nevertheless, some studies were unable to establish a relationship between Type D personality and hypertension. A study of 86 adults without documented cardiovascular disease found that Type D personality was not associated with systolic and diastolic blood pressure [20]. A longitudinal study of a predominantly healthy German working population also failed to find any significant association between Type D personality and components of the metabolic syndrome, including blood pressure [21].

Because Type D classification represents both NA and SI, it is impossible to determine if the association with blood pressure reflects either of these two traits. This study separately explored the two constructs of Type D personality and found that the NA was more strongly associated with blood pressure compared with SI in normotensive adults. NA, the tendency to perceive emotions as negative, could activate the sympathetic–adrenal–medullary system and the hypothalamic–pituitary–adrenal–cortical axis system, which could increase blood pressure through increased serum levels of catecholamines and cortisol [22,23]. Our finding is consistent with a study of 131 healthy young adults which indicated that depression, anxiety, and anger—the main components of NA–were strong predictors of aortic systolic blood pressure, measured by radial artery applanation tonometry [24]. Conversely, a population-based study of 710 middle-aged Finnish men showed no significant direct associations between blood pressure and depressive symptoms, but the two could be mediated through other unfavorable lifestyle variables, such as alcohol consumption, less healthy diet, and inactivity [25]. The relationship between NA and blood pressure clearly warrants additional investigation, particularly using studies with longitudinal design. In addition, whether educational and counselling interventions for individuals with NA can reduce the risk of hypertension will require further studies.

Previous research suggested that emotional dysregulation and coping styles in individual might affect blood pressure [26,27]. Patients with hypertension were found to resort to repression of the emotions generated by a stressful situation [28]. Routine anger suppression was also associated with increased systolic blood pressure response [29]. However, the Z-score transformed SI score was not associated with blood pressure in this study. Whether repression and suppression of emotions are synonymous with the trait SI will require further investigation.

Our study has several limitations. First, this study was based on a relatively small sample size of patients attending general health examination at one regional hospital. Second, because of the cross-sectional design of our study, the temporal relationship between blood pressure and Type D could not be established.

## 5. Conclusions

This cross-sectional study of normotensive Taiwanese patients showed that NA was associated with a higher blood pressure. Assessment of NA may be clinically useful in identifying individuals at risk of hypertension. In addition, research on personality traits, such as forgiveness [30], or reducing NA using mindfulness training [31] may offer new insights that can be used to lower the risk of developing primary hypertension.

## Figures and Tables

**Table 1 medicina-56-00160-t001:** Characteristics of study participants (N = 92).

Variable	Total	Type D	Non-Type D	*p*
	92 (100%)	15 (16.3%)	77 (83.7%)	
Age, mean (SD), years	51.5 (8.1)	47.9 (7.1)	52.2 (8.1)	0.055
Sex, n (%)				0.010
male	40 (43.5)	2 (13.3)	38 (49.4)	
female	52 (56.5)	13 (86.7)	39 (50.6)	
BMI, mean (SD), kg/m^2^	24.4 (3.3)	23.0 (2.4)	24.7 (3.4)	0.068
Educational level, n (%)				0.940
junior high school and below	36 (39.1)	6 (40.0)	30 (39.0)	
senior high school and above	56 (60.9)	9 (60.0)	47 (61.0)	
Marital status, n (%)				>0.999
being married	86 (93.5)	14 (93.3)	72 (93.5)	
other	6 (6.5)	1 (6.7)	5 (6.5)	
Family history of hypertension, n (%)				0.536
yes	29 (33.0)	6 (42.9)	23 (31.1)	
no	59 (67.0)	8 (57.1)	51 (68.9)	
Regular exercise, n (%)				0.219
yes	63 (69.2)	8 (53.3)	55 (72.4)	
no	28 (30.8)	7 (46.7)	21 (27.6)	
Current smoking, n (%)				0.119
yes	15 (16.3)	0 (0)	15 (19.5)	
no	77 (83.7)	15 (100)	62 (80.5)	
Current alcohol use, n (%)				0.288
yes	17 (18.5)	1 (6.7)	16 (20.8)	
no	75 (81.5)	14 (93.3)	61 (79.2)	
Systolic blood pressure, mean (SD), mmHg	119.6 (15.4)	119.9 (18.6)	119.5 (14.8)	0.918
Diastolic blood pressure, mean (SD), mmHg	69.2 (11.1)	70.1 (9.0)	69.0 (11.5)	0.722
Negative affectivity score, mean (SD)	7.66 (6.90)	17.40 (4.08)	5.77 (5.63)	<0.001
Social inhibition score, mean (SD)	7.17 (5.68)	15.60 (4.17)	5.53 (4.32)	<0.001

BMI: body mass index; n: number; SD: standard deviation. There were 4 missing values in family history of hypertension and 1 missing value in exercise. Percentages shown are column percentages except in the header where they are row percentages.

**Table 2 medicina-56-00160-t002:** Simple linear regression analyses of systolic and diastolic blood pressure in normotensive middle-aged and older adults.

Variable	Dependent Variable: Systolic Blood Pressure			Dependent Variable: Diastolic Blood Pressure		
	β	Std. β	*p*	β	Std. β	*p*
Age, years	0.44	0.195	0.025	0.12	0.091	0.390
Sex (male versus female)	2.16	0.070	0.508	3.06	0.138	0.190
Body mass index, kg/m^2^	1.02	0.226	0.046	0.99	0.294	0.004
Educational level (senior high school and above versus below)	−3.33	−0.106	0.314	−3.06	−0.135	0.198
Marital status (being married versus other)	2.73	0.044	0.677	−1.04	−0.023	0.826
Family history of hypertension	0.40	0.013	0.907	4.79	0.215	0.044
Regular exercise	0.80	0.024	0.822	0.09	0.004	0.973
Current smoking	4.52	0.109	0.302	6.06	0.203	0.052
Current alcohol use	1.56	0.039	0.709	2.43	0.086	0.417
Z-score transformed negative affectivity score	2.03	0.132	0.210	1.89	0.170	0.104
Z-score transformed social inhibition score	2.02	0.131	0.214	1.43	0.129	0.220

Std: standardized beta coefficients. Z-score of negative affectivity (NA) was calculated by (NA score–mean NA score)/standard deviation of NA score. Z-score of social inhibition (SI) was calculated by (SI score–mean SI score)/standard deviation of SI score.

**Table 3 medicina-56-00160-t003:** Multiple linear regression analyses of systolic and diastolic blood pressure for Z-score transformed negative affectivity and social inhibition scores in normotensive adults.

Variable	Dependent Variable: Systolic Blood Pressure				Dependent Variable: Diastolic Blood Pressure			
	β	Std. β	*p*	Effect Size	β	Std. β	*p*	Effect Size
Z-score transformed negative affectivity score	3.39	0.220	0.035	0.18	2.38	0.218	0.036	0.28
Z-score transformed social inhibition score	3.02	0.196	0.054	0.17	1.53	0.144	0.154	0.24

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
