# Peer review of "Negative Affectivity Is Associated with a Higher Systolic and Diastolic Blood Pressure in Normotensive Middle-Aged and Older Adults: A Cross-Sectional Study"

_medicina, 2020, doi:10.3390/medicina56040160_

Round 1
Reviewer 1 Report
The authors of this manuscript did a masterful job presenting their findings with regards to Type D personality and how it relates to blood pressure. This new layer and perspective may be clinically useful and provide investigators with a new means of identifying unique populations at risk of have blood pressure abnormalities. Overall the data was presented well and there are no changes to be made.
Author Response
Reviewer 1, Comment No. 1:
The authors of this manuscript did a masterful job presenting their findings with regards to Type D personality and how it relates to blood pressure. This new layer and perspective may be clinically useful and provide investigators with a new means of identifying unique populations at risk of have blood pressure abnormalities. Overall the data was presented well and there are no changes to be made.
Response to Reviewer 1, Comment No. 1:
We greatly appreciate the positive comment from Reviewer 1.
--------------------------------------------------------
Reviewer 2 Report
Summary: The current manuscript investigated whether negative affectivity and social inhibition were associated with systolic and diastolic blood pressure in normotensive middle-aged and older Taiwanese adults. This study evidence that Negative affectivity of the Type D personality was significantly associated with higher systolic and diastolic blood pressure in apparently healthy middle-aged and older adults.
Although the authors present interesting, clinically findings, some aspects could be improved.
Please consider the following suggestion for revision:
Introduction: Overall, the introduction provides a brief background and rationale for the research.
This choice does not allow to analyze the problem fully, and it may be useful to mention other psychological aspects related to blood pressure, such as alexithymia, anxiety, depression, coping strategies. I suggest the reading of the following articles: “Casagrande, M., Forte, G., Guarino, A., Favieri, F., Boncompagni, I., Germanò, R., Germanò, G., Mingarelli, A. (2019). Alexithymia: A Facet of Uncontrolled Hypertension. International Journal of Psychophysiology, 146, 180-189, DOI: 10.1016/j.ijpsycho.2019.09.006” and “Casagrande, M., Boncompagni I., Mingarelli A., Favieri F., Forte G., Germanò R., & Germano G., Guarino, A. (2019) Coping styles in individuals with hypertension of varying severity. Stress & Health, 35, 1-9, DOI: 10.1002/smi.288.”.
Furthermore, the social inhibition and negative affectivity dimensions that define the D-type personality could play a role in other dimensions of health, and their role in cardiovascular health should be clarified.
Methods: The method is succinct, but some improvements should be made to make it more comprehensive. For example, classification in Type D and Non-Type D is not clarified
What is the age range of participants? It would be helpful to report it. Are you sure to cover older age (looking the means and standard deviation is not clear whether older are included)?
Could it have been useful to exclude people over 75, as it could be associated with both D-type personality and vascular changes?
In the description of the instrument to assess Type D personality the NA and SI abbreviation appeared but should be defined before using acronyms
Office blood pressure measurement is in line with current guidelines, or only one measure was provided? It is not clear.
The participant with hypertension levels of blood pressure in-office measurement was excluded?
Analysis: the analyses are well conducted. I appreciate the inclusion of the Effect size of regression. Why did you choose to consider Z-score (conversely to your previous works (rif 15)?
Results: The summary of the stud is well-defined and fits the planned analyses provide. However, in some points, the linguistic style is very confused; a revision would allow us to understand them better.
Discussion: This section could be improved, emphasizing the novelty of this research. It highlights a pattern typical of hypertensive people even in a group of normotensives, and this could be very important from a preventive point of view.
It could be useful to provide a better explanation of the role of emotional regulation or dysregulation.
General comment: I would also encourage the authors to check all references and to proofread the manuscript to improve the English language.
Moreover, the aspect related to aging (middle-aged and older) should be redefined in the light of the overall study, which did unfurnished clearly information on the trend of the relationship of Type-D personality and blood pressure in aging.
Author Response
Reviewer 2, Overall comment:
The current manuscript investigated whether negative affectivity and social inhibition were associated with systolic and diastolic blood pressure in normotensive middle-aged and older Taiwanese adults. This study evidence that Negative affectivity of the Type D personality was significantly associated with higher systolic and diastolic blood pressure in apparently healthy middle-aged and older adults.
Although the authors present interesting, clinically findings, some aspects could be improved.
Response to Reviewer 1, Comment No. 1:
We greatly appreciate the detailed and constructive comments from Reviewer 2.
--------------------------------------------------------
Reviewer 2, Comment No. 1:
Introduction: Overall, the introduction provides a brief background and rationale for the research.
This choice does not allow to analyze the problem fully, and it may be useful to mention other psychological aspects related to blood pressure, such as alexithymia, anxiety, depression, coping strategies. I suggest the reading of the following articles: “Casagrande, M., Forte, G., Guarino, A., Favieri, F., Boncompagni, I., Germanò, R., Germanò, G., Mingarelli, A. (2019). Alexithymia: A Facet of Uncontrolled Hypertension. International Journal of Psychophysiology, 146, 180-189, DOI: 10.1016/j.ijpsycho.2019.09.006” and “Casagrande, M., Boncompagni I., Mingarelli A., Favieri F., Forte G., Germanò R., & Germano G., Guarino, A. (2019) Coping styles in individuals with hypertension of varying severity. Stress & Health, 35, 1-9, DOI: 10.1002/smi.288.”.
Response to Reviewer 2, Comment No. 1:
We greatly appreciate the reviewer for suggesting the two useful articles and we felt that they are excellent for explaining the possible roles of SI. Therefore, we added them in the Discussion section (Line 198).
--------------------------------------------------------
Reviewer 2, Comment No. 2:
Furthermore, the social inhibition and negative affectivity dimensions that define the D-type personality could play a role in other dimensions of health, and their role in cardiovascular health should be clarified.
Response to Reviewer 2, Comment No. 2:
The exact role of the social inhibition and negative affectivity dimensions in cardiovascular health is still an active area of research. Nevertheless, we have added two new references and the following text to the Introduction: “Our previous study showed that both NA and SI were significantly associated with a higher risk of cardiac readmission in both 6 months and 18 months after the initial hospitalization in patients with heart failure [10]. A cross-sectional study on patients with coronary artery disease found that SI and NA were associated with the severity of the disease, which could be explained by the association between SI and poor health behavior as well as a greater stress hormone reactivity associated with NA [11].” (Line 54)
--------------------------------------------------------
Reviewer 2, Comment No. 3:
The method is succinct, but some improvements should be made to make it more comprehensive. For example, classification in Type D and Non-Type D is not clarified.
Response to Reviewer 2, Comment No. 3:
We have added the following sentence in the “2.2. Measurement of Type D Personality” section: A standard cut-off score of ³ 10 on both subscales is used to define Type D personality. (Line 81)
--------------------------------------------------------
Reviewer 2, Comment No. 4:
What is the age range of participants? It would be helpful to report it. Are you sure to cover older age (looking the means and standard deviation is not clear whether older are included)?
Could it have been useful to exclude people over 75, as it could be associated with both D-type personality and vascular changes?
Response to Reviewer 2, Comment No. 4:
We thank the reviewer for this comment. We have included the age range of the participants, which is 40 to 75 years, in the Materials and Methods section, the Results section and the Abstract. We have excluded people over 75 years of age as they could be associated with both D-type personality and vascular changes. (Line 29, 74, 107.)
--------------------------------------------------------
Reviewer 2, Comment No. 5:
In the description of the instrument to assess Type D personality the NA and SI abbreviation appeared but should be defined before using acronyms.
Response to Reviewer 2, Comment No. 5:
The abbreviations NA and SI are defined at their first appearance in the Introduction section (Line 50).
--------------------------------------------------------
Reviewer 2, Comment No. 6:
Office blood pressure measurement is in line with current guidelines, or only one measure was provided? It is not clear.
The participant with hypertension levels of blood pressure in-office measurement was excluded?
Response to Reviewer 2, Comment No. 6:
Only one measure was provided for our blood pressure measurement. The participant with hypertension levels of blood pressure in-office measurement was not excluded. We only excluded patients with self-reported hypertension or those who were currently receiving antihypertensive medication.
--------------------------------------------------------
Reviewer 2, Comment No. 7:
Analysis: the analyses are well conducted. I appreciate the inclusion of the Effect size of regression. Why did you choose to consider Z-score (conversely to your previous works (rif 15)?
Response to Reviewer 2, Comment No. 7:
We used Z-score transformed score in our more recent paper (Lin TK, You KX, Hsu CT, et al. Negative affectivity and social inhibition are associated with increased cardiac readmission in patients with heart failure: A preliminary observation study. PLoS One. 2019;14(4):e0215726.)
Z-score transformed score are easier to interpret when comparing the results between NA and SI.
--------------------------------------------------------
Reviewer 2, Comment No. 8:
Results: The summary of the stud is well-defined and fits the planned analyses provide. However, in some points, the linguistic style is very confused; a revision would allow us to understand them better.
Response to Reviewer 2, Comment No. 8:
We have tried our best to revise the Results section to improve its clarity.
--------------------------------------------------------
Reviewer 2, Comment No. 9:
Discussion: This section could be improved, emphasizing the novelty of this research. It highlights a pattern typical of hypertensive people even in a group of normotensives, and this could be very important from a preventive point of view.
It could be useful to provide a better explanation of the role of emotional regulation or dysregulation.
Response to Reviewer 2, Comment No. 9:
We appreciate the reviewer for this suggestion. We have added the following text to the Discussion and also the reference by Casagrande et al., 2019:
“This is an important finding from a preventive point of view because educational and counselling interventions may be considered in normotensive individuals with Type D personality to reduce the risk of transition to hypertension.” (Line 162)
“Previous research suggested that emotional dysregulation and copying styles in individual might affect blood pressure [26,27]. Patients with hypertension were found to resort to repression of the emotions generated by a stressful situation [28]. In addition, routine anger suppression was associated with increased systolic blood pressure response [29]. However, the Z-score transformed SI score was not associated with blood pressure in this study. Whether repression and suppression of emotions are synonymous with the trait SI will require further investigation.” (Line 197)
--------------------------------------------------------
Reviewer 2, Comment No. 10:
General comment: I would also encourage the authors to check all references and to proofread the manuscript to improve the English language.
Moreover, the aspect related to aging (middle-aged and older) should be redefined in the light of the overall study, which did unfurnished clearly information on the trend of the relationship of Type-D personality and blood pressure in aging.
Response to Reviewer 2, Comment No. 10:
We have checked the references and proofread the manuscript.
We have revised the Abstract, the Materials and Methods section, and the Results section to indicate that the participants were 40 to 75 years of age. (Line 29, 74, 107.)
--------------------------------------------------------
Reviewer 3 Report
Review of “Negative affectivity is associated with a higher systolic and diastolic blood pressure in normotensive middle-aged and older adults: A cross-sectional study” submitted for consideration for publication in Medicina.
This is an interesting, well designed and elaborated study on the relations to the two self-rated subcomponents of Type D personality (negative affectivity and social inhibition) on non-clinical hypertension in middle-aged Taiwanese persons.
The ms is eloquently written to a high standard and can be published with only minor revision as specified in the following in chronological order.
Abstract
Please include an interpretation to the effect sizes reported in the abstract (i.e., medium).
Please justify why apparently healthy persons where investigated – what’s the informational gain as compared to contrasting hypertensive patients with normotensive persons.
Line 59 “has not been reported”
Not YET been reported – or not yet studied (but not reported)?
69 Exclusion criteria included patients with self-reported
70 hypertension or those who were currently receiving antihypertensive medication
Here as well, Please justify why apparently healthy persons where investigated – what’s the informational gain as compared to contrasting hypertensive patients with normotensive persons.
76 The original scale showed good psychometric properties (Cronbach α = 0.88 and
77 0.86; 3-month test-retest reliability = 0.72 and 0.82 for NA and SI, respectively)
Here you must specify to what psychometric properties these scores refer – Is Cronbach alpha for internal reliability? Have test-retest reliabilities also been studied with Cronbach alpha or with correlations?
91 results for NA and SI, their values were transformed INTO Z-scores
Table 1
The indications of % are confusions. In the header, they refer to the entire sample. But in the cells of the table, they refer to the given sub-sample. This must be somewhere explained or maybe the irregular use in the header (referring to the total sample) be omitted and shifted to the foot note. Or indicate in first column.
Lines 112-124
122 transformed SI score was only marginally associated with systolic blood pressure (p = 0.054) and not
123 significantly associated with diastolic blood pressure (p = 0.154) in the regression models.
p-scores do not indicate the magnitude of association. So you cannot state marginally association. What was marginal was the significance. Here and in the entire section from l 112 you should report correlation or regression coefficients, not just the p-values. They depend on sample size and outliers and are therefore uninformative. Therefore, we use effect sizes especially in such small samples as your D-type sample.
Line 155
what is cold stress? Is that a special social term or is that stress due to cold temperature?
Line 167 Germany working
Must be German
181 The relationship between NA and blood pressure clearly warrants additional
182 investigation, particularly studies based on longitudinal design
Not only that – what interventions could become relevant? Educational, counselling, therapeutic?
190 In addition, research on personality traits, such as forgiveness [24] or reducing
191 NA [25] may offer new insights
Reducing NA is not a personality trait, must place a comma after [24]. Here as well do add a few words on what could be studied for reducing NA.
What does it mean that SI was not significant in your view? Why could that be the case?
It was a pleasure to read this well-developed ms. I hope my few comments are helpful.
I don’t have to see this again.
Author Response
Reviewer 3, Comment No. 1:
This is an interesting, well designed and elaborated study on the relations to the two self-rated subcomponents of Type D personality (negative affectivity and social inhibition) on non-clinical hypertension in middle-aged Taiwanese persons.
The ms is eloquently written to a high standard and can be published with only minor revision as specified in the following in chronological order.
Response to Reviewer 3, Comment No. 1:
We highly appreciate the encouraging words from the reviewer and providing helpful comments on improving our manuscript.
--------------------------------------------------------
Reviewer 3, Comment No. 2:
Abstract: Please include an interpretation to the effect sizes reported in the abstract (i.e., medium).
Please justify why apparently healthy persons where investigated – what’s the informational gain as compared to contrasting hypertensive patients with normotensive persons.
Response to Reviewer 3, Comment No. 2:
We have added the effect size in the Conclusion of the Abstract as the following: “Negative affectivity of the Type D personality was significantly associated with higher systolic and diastolic blood pressure, with a medium effect size, in apparently healthy middle-aged and older adults.” (Line 40)
Previous research has already shown that type D personality was found to be significantly associated with hypertension. In our study, we were interested in whether similar association also exists in normotensive individuals.
We have also added the following sentence in the Discussion: “This is an important finding from a preventive point of view because educational and counselling interventions may be considered in normotensive individuals with Type D personality to reduce the risk of transition to hypertension.” (Line 162)
--------------------------------------------------------
Reviewer 3, Comment No. 3:
Line 59 “has not been reported”
Not YET been reported – or not yet studied (but not reported)?
Response to Reviewer 3, Comment No. 3:
We thank the reviewer for asking for clarification of this point. We have revised the sentence to “However, the relationship between the components of Type D personality and blood pressure in normotensive adults has not yet been reported.” (Line 64)
--------------------------------------------------------
Reviewer 3, Comment No. 4:
69 Exclusion criteria included patients with self-reported
70 hypertension or those who were currently receiving antihypertensive medication
Here as well, Please justify why apparently healthy persons where investigated – what’s the informational gain as compared to contrasting hypertensive patients with normotensive persons.
Response to Reviewer 3, Comment No. 4:
Previous research has already shown that type D personality was found to be significantly associated with hypertension. In our study, we were interested in whether similar association exists in normotensive individuals.
--------------------------------------------------------
Reviewer 3, Comment No. 5:
76 The original scale showed good psychometric properties (Cronbach α = 0.88 and
77 0.86; 3-month test-retest reliability = 0.72 and 0.82 for NA and SI, respectively)
Here you must specify to what psychometric properties these scores refer – Is Cronbach alpha for internal reliability? Have test-retest reliabilities also been studied with Cronbach alpha or with correlations?
Response to Reviewer 3, Comment No. 5:
We thank the reviewer for this comment. We have revised the sentence to the following:
“The original scale showed good psychometric properties, including internal reliability with Cronbach alphas of 0.88 for NA and 0.86 for SI and 3-month test-retest reliability with correlation coefficients of 0.72 for NA and 0.82 for SI”. (Line 82)
--------------------------------------------------------
Reviewer 3, Comment No. 6:
91 results for NA and SI, their values were transformed INTO Z-scores.
Response to Reviewer 3, Comment No. 6:
We have corrected the sentence as suggested. (Line 98)
--------------------------------------------------------
Reviewer 3, Comment No. 7:
Table 1: The indications of % are confusions. In the header, they refer to the entire sample. But in the cells of the table, they refer to the given sub-sample. This must be somewhere explained or maybe the irregular use in the header (referring to the total sample) be omitted and shifted to the foot note. Or indicate in first column.
Response to Reviewer 3, Comment No. 7:
We appreciate the reviewer for this comment. We have added the following sentence in the footnote of Table 1: “Percentages shown are column percentages except in the header where they are row percentages.” (Line 117)
--------------------------------------------------------
Reviewer 3, Comment No. 8:
122 transformed SI score was only marginally associated with systolic blood pressure (p = 0.054) and not significantly associated with diastolic blood pressure (p = 0.154) in the regression models.
p-scores do not indicate the magnitude of association. So you cannot state marginally association. What was marginal was the significance. Here and in the entire section from l 112 you should report correlation or regression coefficients, not just the p-values. They depend on sample size and outliers and are therefore uninformative. Therefore, we use effect sizes especially in such small samples as your D-type sample.
Response to Reviewer 3, Comment No. 8:
We thank the reviewer for this comment. We have added standardized beta coefficients to our results. (Line 120-134)
--------------------------------------------------------
Reviewer 3, Comment No. 9:
Line 155: what is cold stress? Is that a special social term or is that stress due to cold temperature?
Response to Reviewer 3, Comment No. 9:
We have revised the term to the exact test, which was the cold pressor test, used in the study by Kupper and colleagues (Line 167).
--------------------------------------------------------
Reviewer 3, Comment No. 10:
Line 167 Germany working
Must be German
Response to Reviewer 3, Comment No. 10:
We have corrected the error (Line 179).
--------------------------------------------------------
Reviewer 3, Comment No. 11:
181 The relationship between NA and blood pressure clearly warrants additional
182 investigation, particularly studies based on longitudinal design
Not only that – what interventions could become relevant? Educational, counselling, therapeutic?
Response to Reviewer 3, Comment No. 11:
We have added the following sentence: “In addition, whether educational and counselling interventions on individuals with NA can reduce the risk of hypertension will require further studies.” (Line 194)
--------------------------------------------------------
Reviewer 2, Comment No. 12:
190 In addition, research on personality traits, such as forgiveness [24] or reducing
191 NA [25] may offer new insights
Reducing NA is not a personality trait, must place a comma after [24]. Here as well do add a few words on what could be studied for reducing NA.
What does it mean that SI was not significant in your view? Why could that be the case?
It was a pleasure to read this well-developed ms. I hope my few comments are helpful.
I don’t have to see this again.
Response to Reviewer 3, Comment No. 12:
Following the reviewer’s suggestion, we have revised the sentence as the following: “In addition, research on personality traits, such as forgiveness [24], or reducing NA using mindfulness training [25] may offer new insights that can be used to lower the risk of developing primary hypertension.” (Line 210)
Regarding the lack of a significant association in SI, we could not offer any possible explanations as little research has been done on elucidating the relationship between SI and blood pressure. However, we have added a new paragraph in the Discussion as the following:
“Previous research suggested that emotional dysregulation and coping styles in individual might affect blood pressure [26,27]. Patients with hypertension were found to resort to repression of the emotions generated by a stressful situation [28]. In addition, routine anger suppression was associated with increased systolic blood pressure response [29]. However, the Z-score transformed SI score was not associated with blood pressure in this study. Whether repression and suppression of emotions are synonymous with the trait SI will require further investigation.” (Line 197)
--------------------------------------------------------
Round 2
Reviewer 2 Report
The paper has been improved, I suggest to accept it in the current version.